# Let's Sample Step by Step:
# Adaptive-Consistency for Efficient Reasoning and Coding with LLMs

**Pranjal Aggarwal[1]    Aman Madaan[3]    Yiming Yang[3]    Mausam[1,2]**

[1]Department of Computer Science, Indian Institute of Technology, Delhi
[2]Yardi School of Artificial Intelligence, Indian Institute of Technology, Delhi
[3]Language Technologies Institute, Carnegie Mellon University

pranjal2041@gmail.com, amadaan@cs.cmu.edu, yiming@cs.cmu.edu, mausam@cse.iitd.ac.in

## Abstract

A popular approach for improving the correctness of output from large language models (LLMs) is Self-Consistency – poll the LLM multiple times and output the most frequent solution. Existing Self-Consistency techniques always generate a *constant* number of samples per question, where a better approach will be to non-uniformly distribute the available budget based on the amount of agreement in the samples generated so far. In response, we introduce Adaptive-Consistency, a cost-efficient, model-agnostic technique that *dynamically* adjusts the number of samples per question using a lightweight stopping criterion. Our experiments over 17 reasoning and code generation datasets and three LLMs demonstrate that Adaptive-Consistency reduces sample budget by up to 7.9 times with an average accuracy drop of less than 0.1%.[1]

## 1 Introduction

The increasing adoption of large language models (LLMs) across various tasks, such as text generation and reasoning (Wei et al., 2022; Kojima et al., 2022; Wang et al., 2022a; Mishra et al., 2022), mathematical reasoning (Lewkowycz et al., 2022; Gao et al., 2022; Arora et al., 2023), and code generation (Li et al., 2022; Madaan et al., 2023b), has underscored the importance of improving the correctness of their outputs. A popular method for achieving this goal is *Self-Consistency* (Wang et al., 2022b), a majority voting technique where multiple output samples are generated for a given input, and the final decision is based on the most frequently occurring output among the samples.

Current Self-Consistency methods typically employ a fixed budget approach, wherein a predetermined number of samples (e.g., 40) are generated to make a decision. However, as LLMs continue to grow in size and complexity, the sampling time and computational costs associated with majority voting become increasingly challenging. This challenge is particularly evident in high-stakes applications like competition-level code generation (Li et al., 2022), where generating a large number of programs, sometimes up to a million, is essential for maximizing performance.

To address this challenge, we introduce *Adaptive-Consistency*, a cost-efficient, model-agnostic majority voting technique. Adaptive-Consistency employs a lightweight stopping criterion that dynamically adjusts the number of samples ($n$) for each input, as opposed to using a fixed budget ($k$). The intuition is that if a clear majority is established with high confidence after sampling fewer than $k$ answers ($n < k$), there is no need to generate additional samples.

Adaptive-Consistency models the probability distribution over unique samples using a Dirichlet distribution, allowing us to quantify the confidence in the lead of the majority element over other elements. For instance, if the majority element has a count of 9 out of the first 10 samples, the likelihood of it remaining the majority element even after 40 samples is very high ($> 99\%$). This allows Adaptive-Consistency to stop sampling at this point, reducing the cost by 30 samples, while Self-Consistency would continue to sample all 40 answers. As an inference-time technique requiring no additional training, Adaptive-Consistency provides a convenient off-the-shelf option for all pre-trained language models, offering the flexibility to balance computational cost and performance.

We evaluate Adaptive-Consistency on 17 diverse tasks and three LLMs of different scales (VICUNA-13B, CODE-DAVINCI-002 and GPT-3.5-TURBO). Our experimental results show that Adaptive-Consistency outperforms Self-Consistency regarding cost efficiency while maintaining comparable output quality. On CODE-DAVINCI-002, Adaptive-

---

[1]Code and LLM outputs are available at https://sample-step-by-step.info/.

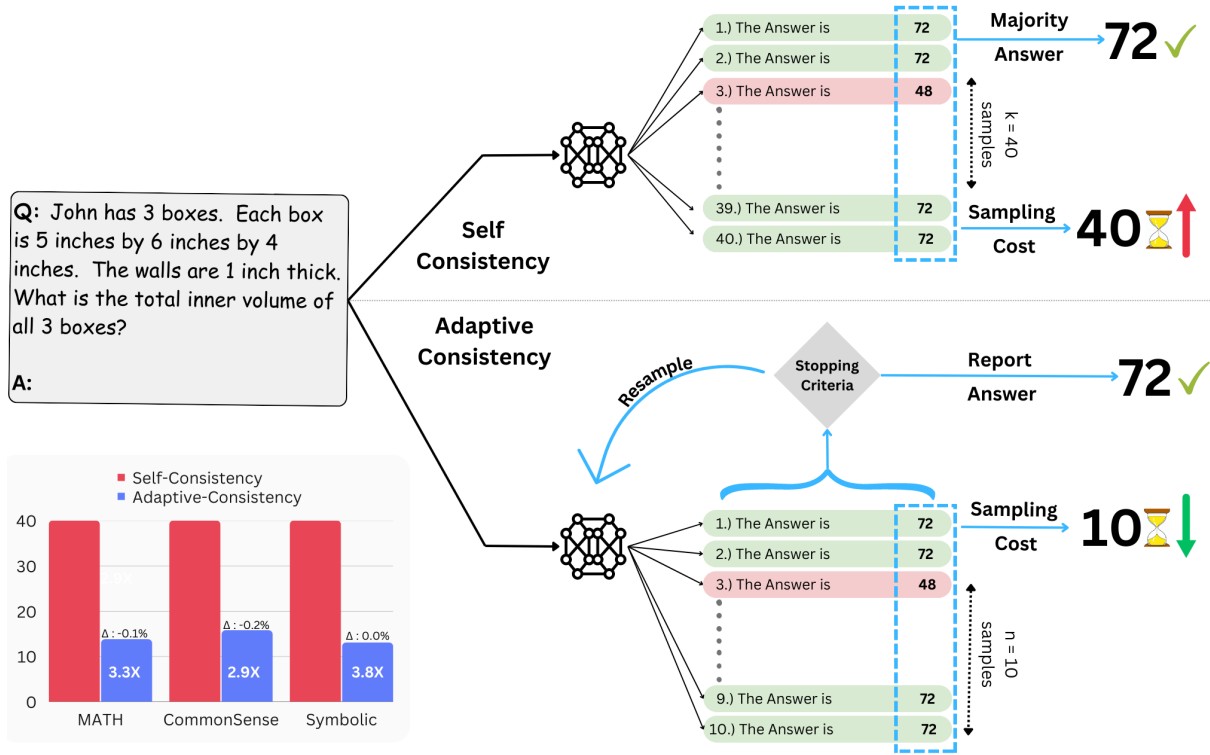

Figure 1: An overview of *Adaptive-Consistency*: Self-Consistency samples a predetermined number of answers, whereas Adaptive-Consistency iteratively samples until a lightweight Stopping Criteria, decides to report the majority answer. The figure demonstrates an example where *Adaptive-Consistency* reduces sampling costs by 4x, requiring only ten samples to report the majority answer. The bottom-left graph contrasts *Adaptive-Consistency* with Self-Consistency across three reasoning categories, showing an average sample budget reduction of $3.3\times$ with a negligible $0.04\%$ drop in accuracy.

Consistency reduces the number of samples required by a factor of $3.4\times$, with no average drop in accuracy. On VICUNA-13B, it requires sampling $1.9\times$ fewer samples, with almost no drop in accuracy. Similarly, on GPT-3.5-TURBO, it samples $4.4\times$ fewer samples, with less than $0.2\%$ drop in accuracy. In summary, our contributions are:

- We propose Adaptive-Consistency, a cost-efficient sampling technique for large language models that dynamically adjusts the number of samples using a lightweight stopping criterion based on the stability of the majority element.

- We conduct extensive experiments using three different LLMs on a diverse set of 17 datasets. These datasets encompass a wide range of tasks, including MATH, COMMONSENSE, SYMBOLIC reasoning, and CODE GENERATION tasks. Adaptive-Consistency consistently and significantly outperforms fixed-budget methods like Self-Consistency, requiring an average of $3.3\times$ fewer samples with less than $0.1\%$ drop in accuracy across all datasets and models.

- Our analysis reveals that for a fixed sampling cost, Adaptive-Consistency consistently achieves better accuracy than Self-Consistency across all datasets (upto 5% absolute points). Additionally, we experiment with various stopping criterias and show the efficacy of Adaptive-Consistency in terms of speed and accuracy.

## 2 Background

**In-Context Few-Shot Prompting** In-context few-shot prompting is a technique employed by large language models (LLMs) to learn and generalize from a limited number of examples provided within the input of a given task. The model can quickly adapt to novel tasks without fine-tuning or additional training by conditioning the model on a few examples. Specifically, a prompt $p$ is constructed by concatenating multiple input-answer example pairs $< x_i, a_i >$. The prompt is then prepended to the test input $x_{test}$, and the model generates the corresponding answer $a_{test}$.

```python
def adaptive_consistency(max_gens,
                         stop_criterion):
    observations = []
    for k in range(1, max_gens):
      observations.append(sample_from_llm())
        if stop_criterion(observations):
            break
    return majority(observations)

def stop_criterion(observations, threshold):
    # Implement your stopping criterion

def self_consistency(max_gens):
    observations = []
    for k in range(1, max_gens):
      observations.append(sample_from_llm())
    return majority(observations)
```

Listing 1: Comparison of Adaptive-Consistency (top) and Self-Consistency (bottom). Self-Consistency always generates a fixed number of samples. In contrast, Adaptive-Consistency uses a lightweight stopping criterion, allowing it to adaptively halt the sampling process, which can lead to improved efficiency and performance.

**Self-Consistency**  Wang et al. (2022b) proposed Self-Consistency which improved performance by sampling multiple diverse reasoning chains and aggregating their outputs using a simple majority voting mechanism. However, higher accuracy is achieved with an increased computational cost, since the LLM must be prompted multiple times for the same question.

## 3  Adaptive-Consistency

Self-Consistency generates a predetermined number of answers ($k$) from the language model (LLM) before returning the majority answer. In contrast, the Adaptive-Consistency method takes an incremental approach to sampling outputs from the language model. After generating each sample, Adaptive-Consistency employs a lightweight *stopping criteria* to determine whether it should 1.) generate an additional sample from LLM or 2.) cease sampling and report the current majority answer. This flexible strategy enables Adaptive-Consistency to dynamically adjust the number of samples generated so far ($n$) for each input. As our experiments demonstrate, $n$ is typically less than $k$ (on average, $3.3\times$ and up to $7.9\times$ less in some cases), allowing Adaptive-Consistency to offer greater cost-efficiency compared to the fixed budget approach employed by Self-Consistency.

Adaptive-Consistency differs from Self-Consistency only in terms of the stopping criteria (Listing 1). The design of the stopping criteria is crucial to our method, as it aims to minimize the average number of samples generated from the LLM while maximizing accuracy. The simplicity of our algorithm allows for the use of various stopping criteria interchangeably, each with its own advantages and disadvantages. We expand on a particular choice of stopping function next.

**Dirichlet Stopping Criteria**  Let $n$ be the number of samples generated from LLM so far, with $m$ unique samples. Let $v = [v_1, v_2, \ldots, v_m]$ be the counts of each element, and $p_i = \frac{v_i}{n}$ be the normalized count. For instance, if $n = 10$, and $m = 3$ (10 samples generated, with 3 unique elements), if $v = [8, 1, 1]$, then we can be more confident that $v_1$ is the answer. On the other hand, if $v = [4, 4, 2]$, then more samples need to be generated. Our goal is to formalize and quantify this intuition.

By convention, let $p_1 = \max(p_i)$. We want to assess the *stability* of $p_1$ as the majority element.[2] Specifically, we want to ask the following question: what is the probability that $p_1$ will be the majority element if we repeat the process of generating $n$ samples again? Intuitively, if this probability is higher than some predetermined threshold $C_{thresh}$, then we can be more confident in our decision to stop sampling and return $p_1$ as the majority element:

$$P(p_1 > \max_{i=2}^{m} p_i \mid v) > C_{thresh}$$

To answer this question, we establish a connection with the Dirichlet distribution. Specifically, we note that the counts $v$ parameterize a Dirichlet distribution, $\text{Dir}(V)$.[3] This connection allows us to explore the behavior of the sampling process by drawing more samples from $\text{Dir}(V)$ and observing the stability of $p_1$ as the majority element. To compute the probability of $p_1$ being the majority element, we can integrate the joint probability density function of the Dirichlet distribution over the appropriate region of the probability simplex. The integral can be expressed as follows:

---

[2]Note that we overload the notation to use $p_1$ to represent both the majority element and its probability (usage clear from context).

[3]Dirichlet is a distribution over multinomials parameterized by counts $V$; each draw from Dirichlet is a multinomial distribution $p$. See Details in Appendix D

$$P(p_1 > \max_{i=2}^{m} p_i \mid V)$$
$$= \int_0^1 \int_{\mathcal{S}(p_1')} f(p_1', p_2, \ldots, p_m \mid V).$$
$$dp_2 \cdots dp_m dp_1',$$

where

$$\mathcal{S}(p_1') = \{(p_2, \ldots, p_m) \mid p_1' > \max_{i=2}^{m} p_i,$$
$$\sum_{i=2}^{m} p_i = 1 - p_1'\}.$$

(1)

In Equation 1, $f(p_1', p_2, ..., p_m|V)$ represents the joint probability density function of the Dirichlet distribution conditioned on the counts $V$. The bounds on the integral for $p_1'$ range from 0 to 1. The probability simplex $\mathcal{S}(p_1')$ is defined for each $p_1'$ value, such that $p_1' > \max_{i=2}^m p_i$, and the remaining $p_i$ values sum to $1 - p_1'$. This constraint ensures that we are considering all possible values of $p_1'$ that would maintain its majority status. Here we assume, that the number of possible unique answers ($m$) is known, based on the current set of observations ($V$). In Analysis ((§ 5.3), we further evaluate a CHINESE RESTAURANT PROCESS (CRP) stopping criteria, which relaxes this assumption by not requiring the number of possible unique answers ($m$) to be known in advance.

**Beta Stopping Criteria** Since the number of unique answers in the observation set can be large, Equation (1) is computationally expensive to solve. As an approximation, we observe that establishing the majority of $p_1$ over the next largest probability, $p_2$, is sufficient for our purpose.

In this setting, the probability in Equation (3) simplifies to a Beta distribution with parameters $(v_1 + 1, v_2 + 1)$, and Equation (1) is replaced by Equation (2). This approximation, which assumes a non-informative prior of BETA$(1, 1)$, allows us to efficiently compute the confidence in $p_1$ being the majority, enabling early stopping decisions without incurring substantial computational overhead.

$$\int_0^{0.5} p_2^{v2} \cdot (1 - p_2)^{v_1} dp_2$$

(2)

Empirically, we show the performance to be similar to Dirichlet stopping criteria but significantly faster (See Section 5.3). Throughout experiments, we refer to this Beta Stopping Criteria as Adaptive-Consistency.

**Code-Generation** We now turn our attention to CODE GENERATION tasks, which involve generating programs that can correctly pass multiple test cases. More details on test case generation can be found in Appendix A.4.

The configuration of code generation tasks significantly impacts the Self-Consistency measurement since different programs might yield varying outputs for a given set of test cases. This variation can cause simple majority voting schemes to be ineffective in evaluating stability. To address this, we explore two distinct methods for aggregating answers across multiple test cases.

In the first method, inspired by the approach used in AlphaCode (Li et al., 2022), we concatenate the outputs for all test cases into a single vector with $t$ elements and apply Self-Consistency across the entire vector. This implies that two programs are considered identical only if their outputs for all $t$ test cases match exactly. However, this simple setup may overestimate the output variance, as different programs can produce distinct outputs for the set of test cases.

To overcome the limitations of the simple setup, we propose an alternative method that treats test inputs as independent entities and applies Adaptive-Consistency to each test case separately:

$$\sqrt[t]{\prod_{j=1}^{t} P(p_1^j > \max_{i=2}^{m} p_i^j \mid V)}$$

(3)

In this equation, $P$ is computed using Equation 1. The Adaptive-Consistency method terminates the sampling process when the normalized probability—expressed as the geometric mean of $P$ across all $t$ test cases—exceeds a predefined threshold (e.g., 0.95).

## 4 Experiments

We evaluate Adaptive-Consistency using 17 diverse benchmark datasets and three different language models. We use prompts by program-aided language models, PAL, (Gao et al., 2022), Self-Consistency (Wang et al., 2022b) and CodeT (Chen et al., 2022).

**Datasets** We evaluate our method on a diverse set of reasoning and coding benchmarks, encompassing 17 datasets across 4 distinct categories:
**1. Mathematical Reasoning:** We use GSM-8K (Cobbe et al., 2021), SVAMP (Patel et al.,

2021), and ASDIV (Miao et al., 2020) which assess the mathematical reasoning capabilities of the LLMs. **2. COMMONSENSE Reasoning Tasks**: We evaluate on 5 datasets: STRATEGYQA (Geva et al., 2021), DATE UNDERSTANDING, SNARKS, RUIN NAMES, SALIENT TRANSLATION that measures different capabilites of LLMs such as multi-hop reasoning and emotional understanding. **3. SYMBOLIC Reasoning Tasks:** We further examine performance on 5 diverse SYMBOLIC reasoning tasks: TRACKING SHUFFLED OBJECTS, LOGICAL DEDUCTION, BOOLEAN EXPRESSIONS, DISAMBIGUATION QA, PENGUINS. **4. CODE GENERATION Tasks** We also evaluate our method on coding tasks, which require to generate a working code given a textual problem description. We evaluate on 4 datasets of varying difficulty: HUMANEVAL (Chen et al., 2021), MBPP (Austin et al., 2021), APPS (Hendrycks et al., 2021) and CODECONTESTS (Li et al., 2022). We refer readers to Appendix A.2 for more details.

**Models**    We evaluate our method on three different language models: **1. GPT-3.5-TURBO:**[4] An RLHF-finetuned GPT-3 based model (unreleased number of parameters). **2. VICUNA-13B:** (Chiang et al., 2023) an open-source transformer model fine-tuned on instruction-following dataset (Taori et al., 2023) from the base Llama series (Touvron et al., 2023). **3. CODE-DAVINCI-002:** A GPT-3-based publicly available model (Brown et al., 2020) which is a part of the Codex series (Chen et al., 2021) and has 175 billion parameters.[5]

**Prompting and Sampling**    We use similar prompts as in PAL (Gao et al., 2022) and CHAIN OF THOUGHT (Wei et al., 2022). Specifically, for mathematical reasoning and DATE UNDERSTANDING tasks, we use prompts from PAL. For other commonsense and SYMBOLIC reasoning tasks, we use CoT (Wei et al., 2022).

For sampling, we follow the scheme suggested in Wang et al. (2022b). Specifically, we use a temperature of 0.7 for sampling and limit the number of generations to a maximum of 40. For coding tasks, we follow the exact procedure as used in CodeT (Chen et al., 2022), with 50 samples for APPS, 100 samples for HUMANEVAL and MBPP

---

[4]https://openai.com/blog/chatgpt

[5]We have access to Codex models through OpenAI's researcher access program. Note that we only need access to the model outputs for this work, and we have released all outputs in the accompanying repository for reproducibility.

and 1000 samples in CODECONTESTS.

**Hyperparameters**    The only hyperparameters in Adaptive-Consistency are those related to parameters in stopping criteria ($C_{thresh}$). We use a high $C_{thresh} = 0.95$ for Adaptive-Consistency. By using a high threshold, we aim to maintain high accuracy and prevent the algorithm from stopping too early. For other Stopping Criteria, we tune parameters on the training set of GSM-8K, and use the same thresholds across all the datasets. The impact of the chosen threshold on the performance of our method is further analyzed in the analysis section (§ 5.1).

**Baselines**    We compare our method against Self-Consistency, which is the current state-of-the-art method. Further, in Section 5.3, we evaluate Adaptive-Consistency against different stopping criteria, such as RANDOM stopping and MAJORITY (stopping at majority), ENTROPY, DIRICHLET and CRP.

**Evaluation Metrics**    We evaluate the performance of our method and the baselines using two metrics: average generations sampled from the LLMs, and overall reasoning accuracy. Our results show that Adaptive-Consistency achieves similar performance to Self-Consistency while often reducing sample budget considerably.

### 4.1 Results

Table 1 presents the main results, and is divided into two parts showing results across different task categories (top sub-table) and on various language models (bottom sub-table). We focus on the potential tradeoff between efficiency and accuracy.

**Results Across Task Categories**    Our experimental results demonstrate the significant efficiency gains achieved by Adaptive-Consistency across different task categories – $3.3\times$ times fewer samples in mathematical tasks with a 0.1% accuracy drop, $2.9\times$ times fewer samples in commonsense tasks with a 0.2% accuracy drop, $3.8\times$ times fewer samples in symbolic reasoning tasks maintaining accuracy, and $2.4\times$ times fewer samples in coding tasks while improving accuracy by 0.4%. These findings confirm the effectiveness of Adaptive-Consistency in identifying the majority element early, highlighting its potential across various applications, including reasoning and coding.

| Category | Accuracy | | Num. Generations | | $\Delta$ | |
| --- | --- | --- | --- | --- | --- | --- |
| | Self-Consistency | Adaptive-Consistency | Self-Consistency | Adaptive-Consistency | Num. Gen. | Acc. |
| MATH | **73.2** | 73.1 | 40 | **13.8** | 3.3× | -0.1 |
| COMMONSENSE | **66.0** | 65.8 | 40 | **15.8** | 2.9× | -0.2 |
| SYMBOLIC Reasoning | 72.8 | 72.8 | 40 | **13.1** | 3.8× | +0.0 |
| CODE GENERATION | 35.2 | **35.6** | 312.5 | **173.6** | 2.4× | +0.4 |

| Model | Accuracy | | Num. Generations | | $\Delta$ | |
| --- | --- | --- | --- | --- | --- | --- |
| | Self-Consistency | Adaptive-Consistency | Self-Consistency | Adaptive-Consistency | Num. Gen. | Acc. |
| GPT-3.5-TURBO | **76.4** | 76.2 | 40 | **10.0** | 4.4× | -0.2 |
| VICUNA-13B | 54.0 | **54.1** | 40 | **21.7** | 1.9× | +0.0 |
| CODE-DAVINCI-002 | 69.7 | **69.8** | 104.1 | **49.4** | 3.4× | +0.0 |

Table 1: **Main results:** Adaptive-Consistency achieves a significant reduction in the number of generations, with a negligible impact on accuracy. The $\Delta$ columns display reductions in generations (Num. Gen.) and accuracy (Acc.) between Self-Consistency and Adaptive-Consistency. Detailed results are in Table 5.

**Results Across Language Models** Examining the results across different language models, we find that Adaptive-Consistency is model-agnostic, and consistently reduces the number of generations with minimal to no impact on accuracy. Adaptive-Consistency consistently reduces the number of generations required, with reductions of 4.4× for GPT-3.5-TURBO, 1.9× for VICUNA-13B, and 3.4× for CODE-DAVINCI-002, highlighting its cost-effective nature and adaptability to different scales of models. Moreover, the minimal accuracy differences and slight improvements showcase the practical utility of Adaptive-Consistency, emphasizing its diverse applicability and model-agnostic characteristics.

## 5 Analysis

### 5.1 Effect of Confidence Threshold in Adaptive-Consistency

The confidence threshold, $C_{thresh}$, is a crucial hyperparameter for Adaptive-Consistency, as it determines when to stop sampling based on the desired level of confidence in the majority element. While we set the threshold to a stringent value of 0.95 for all experiments, in this section, we analyze the impact of varying $C_{thresh}$ from 0.5 to 1 to understand the trade-offs between model accuracy and cost-efficiency.

In Figure 2, we present a visualization that examines the relationship between the confidence threshold, $C_{thresh}$, and the performance of adaptive consistency in terms of both accuracy and cost-efficiency. The x-axis represents the confidence threshold, varying from 0.5 to 1. The left y-axis

displays the model's accuracy, while the right y-axis shows the average number of samples drawn.

The plot (for GSM-8K) shows the expected behavior of two curves: the blue curve (accuracy) increases gradually and then plateaus, while the red curve (average number of samples) initially increases linearly and then climbs more steeply. The plateau in accuracy signifies that the model has reached its maximum achievable accuracy, and further sampling will not improve it much. Meanwhile, the red curve's climbing rate indicates that the model requires more samples to meet an increasingly stringent confidence threshold for stopping, highlighting the trade-off between accuracy and cost efficiency. We refer readers to Appendix C.4 for more results.

### 5.2 Adaptive-Consistency vs. Self-Consistency For Equal Average Sample Costs

Section 4.1 previously demonstrated that Adaptive-Consistency achieves comparable performance to Self-Consistency using fewer samples. In this section, our primary objective is to compare the performance of Adaptive-Consistency to Self-Consistency across various sampling budgets. For each fixed sampling budget $k$, we contrast the performances of Adaptive-Consistency and Self-Consistency, where Self-Consistency distributes sample budget uniformly to each question, Adaptive-Consistency uses nonuniform allocation, rather than consistently across all instances.

We evaluate Adaptive-Consistency using varying thresholds, with each threshold producing a distinct point (#samples, performance) on the cost-quality curve. For every specific sample count

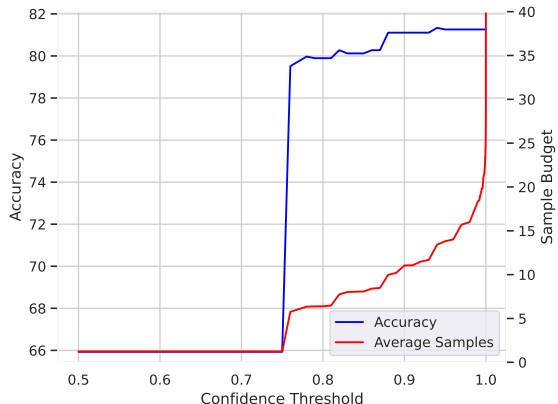

| | BETA | ENTROPY | DIRICHLET | CRP |
|---|---|---|---|---|
| Time (ms) | 0.03 | 0.03 | 101.3 | 94.6 |

Table 2: Time comparison of different stopping criterias, when evaluated on GSM-8K and DATE UNDERSTANDING datasets. All stopping criterias are significantly faster than LLM inference, with BETA being 4 orders of magnitude faster than other variants.

Figure 2: Impact of Confidence Threshold ($C_{thresh}$) on Adaptive-Consistency for GSM-8K: As $C_{thresh}$ varies, the accuracy of Adaptive-Consistency increases gradually, eventually plateauing. Initially, the average number of generations also increases gradually but then sharply climbs, reflecting the accuracy-confidence trade-off.

(#samples) generated by Adaptive-Consistency, we subsequently run Self-Consistency to obtain its corresponding performance. The relationship between the two methods across these data points is visualized in Figure 3 which provides a visual comparison of the performance of Adaptive-Consistency and Self-Consistency on GSM-8K. Adaptive-Consistency outperforms Self-Consistency in accuracy across all average sample costs. For example, when the average sample cost is 10, Adaptive-Consistency achieves approximately 3% higher accuracy on GSM-8K. Similar results hold on other datasets; see Appendix C.1 for full results.

The success of Adaptive-Consistency can be attributed to the fact that it varies the number of samples based on the complexity of the instance, using more samples where a clear consensus is hard to reach and fewer where answers are consistent. Consequently, Adaptive-Consistency achieves improved overall performance when controlled for cost budget.

### 5.3 Evaluation of Different Stopping Functions

Adaptive-Consistency allows a flexible choice of stopping criteria, based on intended objective and requirements. Here, we evaluate six different functions: 1) RANDOM: randomly stopping with a probability $p$, 2) MAJORITY: stopping after the most common answer has a majority above a threshold, 3) ENTROPY: stopping after the entropy of answers is below a threshold, 4) BETA: The main stopping criterion used in Adaptive-Consistency, based on Equation (2), 5) DIRICHLET: The stopping criterion, based on Equation (1), 6) CHINESE RESTAURANT PROCESS (CRP): Unlike DIRICHLET, CRP makes no assumption on the number of possible unique answers. Based on the available observations, we first model the concentration parameter ($\alpha$), denoting the probability of getting a new answer, then perform Monte Carlo simulations to obtain stability of the current majority (see Appendix C.3 for more details).

The parameters for all these methods are tuned, as discussed in Section 4. Figure 4 compares BETA to ENTROPY and MAJORITY over a range of expected sampling costs. BETA consistently achieves higher accuracy than both for the same sampling cost. Further, we find RANDOM to be the least effective method as expected, whereas MAJORITY almost consistently underperforms both BETA and ENTROPY. While DIRICHLET and CRP have a similar performance to BETA, they are both about four orders of magnitude slower than BETA due to the expensive multivariate integral calculation. Nonetheless, despite being run on a single cpu core, even DIRICHLET and CRP have negligible time and cost compared to LLM inference. The exact timings are presented in Table 2. The detailed results are presented in Appendix C.2, Table 7.

In summary, Adaptive-Consistency is particularly effective in two scenarios: *(i)* when a majority trend is evident early in the sampling process, such as in the SVAMP dataset where it achieves comparable accuracy to Self-Consistency using fewer than 5 samples on average per input; and *(ii)* for tasks with a limited set of potential answers, such as the BOOLEAN EXPRESSIONS dataset where Adaptive-Consistency reduces the computational budget by 7.9 times without any loss in accuracy.

## 6 Related Work

**Crowdsourcing and Adaptive Consistency**
*Adaptive-Consistency* finds inspiration in tech-

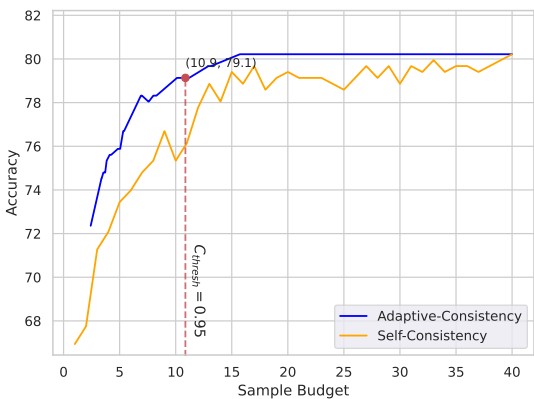

Figure 3: Comparison of Adaptive-Consistency with Self-Consistency on various average sampling costs on 2 datasets: GSM-8K and DATE UNDERSTANDING. Adaptive-Consistency is able to consistently beat Self-Consistency, especially when the sampling cost is low.

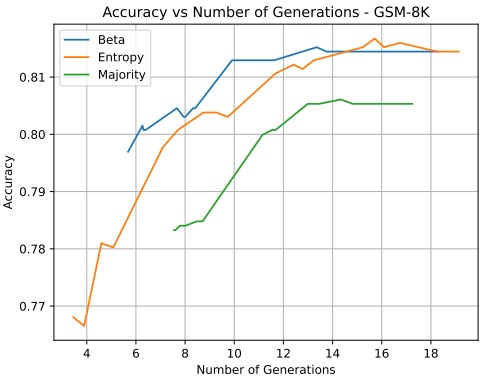

Figure 4: Comparison of BETA, ENTROPY and MA-JORITY stopping criterias. BETA consistently beats EN-TROPY and MAJORITY in terms of accuracy for the same sampling cost.

niques from crowdsourcing (Lin et al., 2012; Dai et al., 2013; Weld et al., 2015; Bragg et al., 2016). Traditionally, crowdsourcing involves aggregating diverse human judgments, which presents challenges in managing resource allocation—knowing when to query additional contributors or stop based on the consistency of responses (Doan et al., 2011; Quinn and Bederson, 2011). Early research concentrated on probabilistic models estimating the 'true' answer and worker reliability (Dawid and Skene, 1979; Whitehill et al., 2009), later considering factors like worker expertise, task complexity, and answer quality (Raykar et al., 2010; Welinder et al., 2010). However, rather than addressing these issues with multiple human contributors, *Adaptive-Consistency* is tailored specifically for LLMs, optimizing for computational efficiency and output

accuracy. In line with our vision, (Parameswaran et al., 2023) have recently proposed declarative prompt engineering, viewing LLMs like crowd workers and leveraging multiple prompting strategies.

**Architectures for adaptive computation** A related body of work on adaptive computation aims to preempt computation based on intermediate representations (Liu et al., 2020; Zhou et al., 2020; Schuster et al., 2021; Geng et al., 2021; Xin et al., 2020). Schuster et al. (2022) present CLAM, a language model that performs language generation adaptively. Hou et al. (2020) propose Dynamic Bert, which can adapt the depth and width of the transformer to satisfy various computational constraints. Xing et al. (2020) propose a dynamic deep neural network with an early-exit strategy embedded for enhancing the quality of compressed images. Another direction of work focuses on pruning model weights or training sparse weights (Fan et al., 2019; Jayakumar et al., 2021) to reduce training and inference time. In contrast to these methods, our approach completely obviates making any architectural modifications.

**Inference-time adaptive computation** These methods focus on adaptive computation at inference time without making architectural modifications to the models. Schwarzschild et al. (2021b,a) focus on three different generalization tasks. They observe that increasing the number of test iterations (which corresponds to the network depth in their setting) helps the models in generalizing better to difficult problems. Madaan and Yang (2022) leverage two different networks trained for the same task, a larger variant (slow) and a smaller variant (fast). The switch from fast to slow happens during inference, based on the complexity of generation at the current step. Xue et al. (2023) train language models to adaptively read tokens from a tape bank for each input. Different from these works, our focus is tasks where the multiple samples are drawn from a model (vs. iteratively solving a task, which is a focus of these works). Additionally, recent works such as (Madaan et al., 2023a; Chen et al., 2023) have propsed to adaptively selecting models of varying sizes based on verification signals derived from the output of the smaller model. Our methods, however, distinguish themselves by not necessitating the use of an additional verifier, and without the need of multiple models.

**Adaptive Sampling in Training and Active Learning** Another line of work focuses on importance-based sampling of input instances during training (Bengio and Senecal, 2008; Prabhu et al., 2019; Berger et al., 2017). In contrast to the aforementioned methods, our approach centers on adaptively sampling multiple outputs per input instance during the inference phase, without soliciting additional labels. Our method is crafted to efficiently obtain reliable predictions from pretrained language models by adaptively sampling their outputs, distinguishing it from both adaptive sampling in training and active learning, which focus on the training phase.

## 7 Conclusion and Future Work

This paper presented Adaptive-Consistency, a cost-efficient and model-agnostic technique for improving the correctness of output from large language models (LLMs) using dynamic sampling. Our approach builds upon the Self-Consistency method and introduces a lightweight stopping criterion that allows for adaptive sampling based on the amount of agreement in the samples drawn so far. Adaptive-Consistency is effective across 17 datasets and three LLMs, on both reasoning and coding tasks. It reduces the required sample budget by 2 to 4 times, while maintaining comparable accuracy, with an average drop of less than 0.1%.

Our work opens up several avenues for future research. We may develop alternative stopping criteria, or combining multiple criteria could lead to even more efficient sampling techniques. Moreover, in our current approach, the majority decision relies on using matches to determine the most common answer. However, this may not always capture the true majority, e.g., in generative tasks, where the output can have variations that do not affect the overall correctness or relevance of the answer. To foster further research and enable reproducibility, we have released the code and LLM outputs at https://sample-step-by-step.info/.

## Acknowledgements

We thank the anonymous reviewers for their useful comments and suggestions. Mausam is supported by grants from Microsoft, Google and Verisk, Wipro CoE on generative AI, Yardi School of AI travel funds, and the Jai Gupta chair fellowship by IIT Delhi. We thank the IIT Delhi HPC facility for its computational resources. This work was also partially supported by the CSE Research Acceleration Fund of IIT Delhi Aman is supported by a contract from the DARPA KAIROS program under agreement number FA8750-19-2-0200. The U.S. Government is authorized to reproduce and distribute reprints for Governmental purposes, notwithstanding any copyright notation thereon. The views and conclusions contained herein are those of the authors and should not be interpreted as necessarily representing the official policies or endorsements, either expressed or implied, of the U.S. Government.

## Limitations

Despite the promising results of our proposed Adaptive-Consistency method, it bears several limitations and scopes for future improvement.

- **Stopping criterion sensitivity:** The current stopping criterion, based on the majority element's stability in the sample set, may not always indicate sample agreement optimally. Instances may arise where the majority element lacks stability, yet the criterion triggers, potentially leading to suboptimal decisions. Future work could explore more robust or alternative stopping criteria.

- **Generalizability:** The effectiveness of our method may vary across tasks or models, despite testing on a diverse range of 17 datasets and three different LLMs of contrastive scale. Notably, Adaptive-Consistency is anticipated to fail where Self-Consistency fails.

- **Task-specific adaptations:** The task-agnostic nature of Adaptive-Consistency might limit its performance on tasks that could benefit from task-specific adaptations. Specialized versions of Adaptive-Consistency for specific tasks or domains could potentially enhance performance. We have initiated this by experimenting on CODE GENERATION dataset, but extending Adaptive-Consistency to other domains may not be as straightforward.

- **Reliance on the pretrained LLM:** Our method depends on the pretrained LLM for generating multiple samples. Consequently, any limitations or biases in the LLM would persist in the Adaptive-Consistency. Addressing these issues might require improvements in the LLM training process itself or the integration of external knowledge sources.

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

## A Experimental Setup

### A.1 Hyperparameters

The only hyperparameters in Adaptive-Consistency, are those related to parameters in stopping criterias ($C_{thresh}$). We use a high $C_{thresh} = 0.95$ for Adaptive-Consistency. By using a high threshold, we aim to maintain high accuracy and prevent the algorithm from stopping too early. For other Stopping Criterias, we tune our parameters on the training set of GSM-8K, and use the same thresholds across all the datasets. The impact of the chosen threshold on the performance of our method is further analyzed in the Analysis Section (§ 5.1). We further evaluate all methods on a set of 3 seeds and report the table with standard deviation in Table 5. We use only a single seed for GPT-3.5-TURBO because of the cost associated.

### A.2 Benchmarks

We evaluate our method on a diverse set of coding and reasoning benchmark datasets, encompassing 17 datasets across four distinct categories:

1. **MATHEMATICAL Reasoning:** To assess mathematical reasoning capabilities, we utilize the following datasets: GSM-8K (Cobbe et al., 2021), SVAMP (Patel et al., 2021), and ASDIV (Miao et al., 2020). These datasets consist of grade-school-level algebra word problems necessitating arithmetic operations and problem-solving based on contextual information.

2. **COMMONSENSE Reasoning Tasks:** We evaluate Adaptive-Consistency on four COMMONSENSE reasoning tasks. **1.) STRATEGYQA** (Geva et al., 2021) comprises questions that demand the model to infer a multi-hop strategy with reasoning steps implicitly embedded in the questions. **2.) DATE UNDERSTANDING** entails questions that require the model to deduce dates from natural language descriptions and perform arithmetic operations accordingly. **3.) SALIENT TRANSLATION** is a salient translation error detection task that requires the model to identify the type of error in a translation. **4.) SNARKS** and **5.) RUIN NAMES** both focus on emotional understanding tasks.

3. **SYMBOLIC Reasoning Tasks:** We examine the performance of our method on six diverse SYMBOLIC reasoning tasks. **1.) TRACKING SHUFFLED OBJECTS** is a tracking task that necessitates the model to infer the final state of a system, given its initial state and a sequence of modifications. **2.) LOGICAL DEDUCTION** is a logical deduction

| Dataset | $|N\_test|$ | Answer Format |
|---|---|---|
| GSM-8K | 1319 | Numerical |
| ASDIV | 2096 | Numerical |
| SVAMP | 1000 | Numerical |
| DATE UNDERSTANDING | 369 | String |
| TRACKING SHUFFLED OBJECTS | 250 | MCQ |
| LOGICAL DEDUCTION | 250 | MCQ |
| STRATEGYQA | 2279 | Binary |
| BOOLEAN EXPRESSIONS | 250 | Binary |
| SNARKS | 250 | Binary |
| RUIN NAMES | 178 | MCQ |
| SALIENT TRANSLATION | 250 | MCQ |
| DISAMBIGUATION QA | 250 | MCQ |
| PENGUINS | 146 | MCQ |
| HUMANEVAL | 164 | Python Code |
| MBPP | 427 | Python Code |
| APPS | 5000 | Coding |
| CODECONTESTS | 165 | Competitive Coding |

Table 3: Dataset Statistics. We evaluate on 17 diverse reasoning datasets, having different difficulty, domains, answer types, sizes ($N\_test$).

task that demands the model to deduce the order of a sequence of objects based on a minimal set of conditions. **3.) BOOLEAN EXPRESSIONS** is a boolean expressions task that evaluates whether a language model has learned the rules of deductive reasoning, i.e., formal (zeroth-order) logic associated with the words "and," "or," "not," etc. **4.) DISAMBIGUATION QA** is a disambiguation task that necessitates the model to select the person to whom the pronoun refers. **5.) PENGUINS** describes a table of penguins and requires the model to answer questions about the penguins' attributes.

4. **CODE GENERATION Tasks:** We further evaluate the performance of our method by conducting experiments on four diverse standard coding tasks. These tasks encompass a range of programming challenges, including both basic human-written and crowd-sourced Python tasks found in the **1.) HUMANEVAL** (Chen et al., 2021) and **2.) MBPP** (Austin et al., 2021) datasets, as well as more challenging competition-level coding tasks from the **3.) APPS** (Hendrycks et al., 2021) and **4.) CODECONTESTS** (Li et al., 2022) datasets.

### A.3 Tools and Framework

For querying GPT-3.5-TURBO and CODE-DAVINCI-002 models (Chen et al., 2021), we use the api library provided by OpenAI[6]. We use the official code provided for running VICUNA-13B model (Chiang et al., 2023). We run inference on VICUNA-13B models on single A100 gpus. For coding tasks, we use the outputs provided by CodeT (Chen et al., 2022), where models are zero-shot prompted with temperature=0.8, and top_p =

---

[6]API available at: https://platform.openai.com/

0.95. stopping criteria in Adaptive-Consistency are fast to run, and we use a single-core machine. For numerical integration, we use the Scipy library in Python.

### A.4 Test-Case Generation

For CODE GENERATION tasks, we generate test cases in a similar fashion to CodeT (Chen et al., 2022). Specifically, we prompt the model with function description and prompt for generation of assert statements. However, unlike CodeT, we limit ourselves to only 10 test cases, which are generated in 1-2 prompts to LLM, thus adding neglible effect on the code generation itself.

Dataset Statistics are presented in Table 3.

## B  Results

We present the complete results with standard deviation in Table 5. For CODE GENERATION tasks, results are presented in Table 4

Further in Table 6 we show that improvements by Adaptive-Consistency are statistically significant across all datasets. We perform 2 sample t-test on 3 random seeds. While p-value of number of generations is much less than 0.05 (average: 1.5e-3), indicating that our method is significantly more efficient, the p-value of accuracy is much larger than 0.05 (average: 0.50), indicating that the slight accuracy difference between baseline and our method is statistically insignificant.

## C  Analysis

### C.1  Adaptive-Consistency vs. Self-Consistency For Equal Average Sample Costs

In Section 5.2, we demonstrate that Adaptive-Consistency achieve better accuracy over Self-Consistency when both are operating on same expected sample cost. In Figure 5 we show the complete results.

Section 4.1 previously demonstrated that Adaptive-Consistency achieves comparable performance to Self-Consistency using fewer samples. In this section, we consider a scenario where Adaptive-Consistency and Self-Consistency operate with the same average number of samples. For each fixed sampling budget $k$ of Self-Consistency, we contrast the performance of Adaptive-Consistency and Self-Consistency, where Adaptive-Consistency uses $k$ samples on average, rather than consistently across all instances.

Figure 3 provides a visual comparison of the performance of Adaptive-Consistency and Self-Consistency on GSM-8K: Adaptive-Consistency outperforms Self-Consistency in accuracy across all average sample costs. For example, when the average sample cost is 10, Adaptive-Consistency achieves approximately 3% higher accuracy on GSM-8K.

The success of Adaptive-Consistency can be attributed to its adaptive sampling strategy. By varying the number of samples based on the complexity of the instance—using more samples where a clear consensus is hard to reach and fewer where answers are consistent—Adaptive-Consistency manages to secure improved overall performance even when the average sample cost matches that of Self-Consistency.

### C.2  Stopping Criterias

This section follows from the main discussion in Section 5.3. We evaluate different stopping criterias for Adaptive-Consistency. We evaluate 6 different functions:

1. RANDOM: randomly stopping with a probability $p$, 2.)

2. MAJORITY: stopping after the most common answer has a majority above a threshold,

3. ENTROPY: stopping after the entropy of answers is below a threshold,

4. BETA: The main stopping criteria used in Adaptive-Consistency, based on the Equation (2),

5. DIRICHLET: The stopping criteria, based on Equation (1).

6. CHINESE RESTAURANT PROCESS (CRP): The stopping criteria, which models probability as chinese restaurant process making no assumption on possible number of unique answers.

For comparison, we tune the $C_{thresh}$ in each case on the training set of GSM-8K dataset. Results are presented in Table 7. RANDOM and MAJORITY are inferior to BETA across all datasets and models. Further, while DIRICHLET and CRP are almost similar to BETA, they are relatively very slow. While Although, from Table 7, ENTROPY looks appears to be on par with BETA, in Figure 6, we show BETA

| | Model | Self-Consistency | | Adaptive-Consistency | | $\Delta$ | |
|---|---|---|---|---|---|---|---|
| | | Avg. Gen. | Accuracy | Avg. Gen. | Accuracy | Gen. Reduc. | Acc. Diff. ↑ |
| **HUMANEVAL** | CODE-DAVINCI-002 | 100 | 61.4 | **23.6** | **63.4** | 4.3× | +2.0 |
| | INCODER-6B | 100 | 19.5 | **51.2** | **20.1** | 2.0× | +0.6 |
| | CODEGEN-16B | 100 | 34.1 | **54.7** | **36.0** | 1.8× | +1.9 |
| **MBPP** | CODE-DAVINCI-002 | 100 | **64.4** | **36.3** | 63.9 | 2.8× | −0.5 |
| | INCODER-6B | 100 | 30.7 | **53.8** | **30.9** | 1.9× | +0.2 |
| | CODEGEN-16B | 100 | 49.6 | **57.8** | **50.4** | 1.7× | +0.8 |
| **APPS** | CODE-DAVINCI-002 | 50 | **11.9** | **44.4** | 11.9 | 1.1× | 0.0 |
| **CODECONTESTS** | CODE-DAVINCI-002 | 1000 | **3.0** | **590.2** | 3.0 | 1.6× | 0.0 |

Table 4: Comparison of Adaptive-Consistency with Self-Consistency on 4 diverse code generation datasets. The table presents the accuracy of Self-Consistency, the average number of generations (Avg. Gen.) for Adaptive-Consistency, and the accuracy of Adaptive-Consistency. Self-Consistency always draws a fixed number of samples. The $\Delta$ columns display the reduction in generations (Gen. Reduc.) and the difference in accuracy (Acc. Diff.) between Self-Consistency and Adaptive-Consistency. For CODECONTESTS, Self-Consistency uses 1000, APPS use 50, while HUMANEVAL and MBPP use 100 generations each.

beats ENTROPY given the same expected sampling cost.

Finally, BETA has additional key advantages: BETA incorporates a measure of uncertainty, which makes it more robust to variations in data order, mitigates the influence of noise, and offers a quantitative measure of confidence in the majority outcome. Consider an extreme case where the first two generated solutions are identical. The majority voting strategy would instantly halt the process, potentially missing out on better solutions. In contrast, BETA will keep sampling as the confidence for stopping has not yet reached.

### C.3 Chinese Restaurant Process

In the DIRICHLET stopping criteria, we assume that the number of unique answers that can be generated by the LLM is known in advance (and equal to the number of unique answers in the current observation set). However, this assumption may not hold for datasets such as GSM-8K, where numerical answers are expected. The CHINESE RESTAURANT PROCESS (CRP) is a generalization of the DIRICHLET process that addresses this limitation by not making any assumption on the number of unique answers.

In CRP, we consider a list of same answers as a cluster, denoted by $c_i$, where $i$ is the index of the cluster. Let $n_i$ be the number of elements in cluster $c_i$, and $n$ be the total number of elements across all clusters. The probability of a new answer belonging to an existing cluster $c_i$ is directly proportional to the size of the cluster, and is given by:

$$P(c_i) = \frac{n_i}{n + \alpha}, \quad (4)$$

whereas the probability that a new unseen answer will form a new cluster is given by:

$$P(c_{new}) = \frac{\alpha}{n + \alpha}, \quad (5)$$

where $\alpha$ is the concentration parameter, which parameterizes the probability of generating a new answer.

Our goal is to calculate the probability that the current majority cluster in observations will remain the same even with more generations. The first task is to estimate the concentration parameter $\alpha$. We use the approximation proposed by (West, 1992) to model the $\alpha$ as

$$p(\alpha|k, n) \approx G(a + k - 1, b + \gamma + \log(n)), \quad (6)$$

where $k$ is the number of unique answers (clusters) in the current observation, $n$ is the total number of answers, $a$ and $b$ are priors and both set equal to 1, and $\gamma$ is Euler's constant and $G(\alpha; a + k - 1, b + \gamma + \log(n))$ denotes the probability density function of the Gamma distribution with shape parameter $a + k - 1$ and rate parameter $b + \gamma + \log(n)$.

We sample $\alpha$ multiple times (100), and for each sample, we run Monte-Carlo Simulation (1000 simulations) based on the CRP probability modeling.

Each simulation starts from from current set of observations, and performed till 40 generations are sampled. The probability that the current majority cluster remains the majority is then given by:

$$P(\text{majority}) = \frac{1}{N_\alpha N_{MCS}} \sum_{i=1}^{N_\alpha} \sum_{j=1}^{N_{MCS}} I(\text{majority}_n^{40}), \tag{7}$$

where $N_\alpha$ is the number of times we sample $\alpha$, $N_{MCS}$ is the number of Monte-Carlo Simulations, and $I(\text{majority}_n^{40})$ is an indicator function that equals 1 if the current majority remains the majority after 40 generations, and 0 otherwise.

### C.4  Effect of Confidence Threshold on Adaptive-Consistency

We follow the discussion in Section 5.1, and present complete results on all datasets for CODE-DAVINCI-002.

## D  Derivation of DIRICHLET stopping criteria

Consider for a given input ($I$), the model can generate one of $m$ distinct answers $A := \{a_1, a_2, \ldots a_m\}$. Define the probability of generating an answer given input as $p_i := P(a_i \mid I)$. Now, consider an observation set ($O$) with counts of each of $a_i$ as $v_i$, such that $\sum_{i=1}^m v_i = n$. Now, without loss of generality, consider $p_1 > \max_{i=2}^m p_i$. Now, based on Equation (3), we need to find the probability:

$$P(p_1 > \max_{i=2}^m p_i \mid O)$$

.

However, here the $p_i$s are latent variables, and only $O$ is available to us. We next make the following

**Assumption 1:** The vector $\vec{p} = \{p_1, p_2 \ldots p_m\}$ is sampled from uniform distribution over $(m-1)$-simplex.

Thus, $p_1 = 1 - \sum_{i=1}^{m-1} p_i$. Since the observation set follows a multinomial distribution with parameters $\vec{p}$, conditional joint probability distribution of $O$ given $\vec{p}$ can be written as:

$$P(O \mid \vec{p}) = \frac{n!}{\prod_{i=1}^m (v_i!)} \prod_{i=1}^m p_i^{v_i} = Dir(v_1+1, v_2+1 \ldots v_m+1)$$

, where $Dir$ represents the dirichlet distribution with $v_i + 1$, as its parameters. Applying Baye's

Rule,

$$P(\vec{p} \mid O) = \frac{P(O \mid \vec{p}) \cdot P(\vec{p})}{P(O)}$$

. Here $P(O)$ is a normalizing constant and can be omitted for computation. From Assumption 1, since $\vec{p}$ is sampled from uniform distribution,

$$P(\vec{p}) = \prod_{i=2}^m dp_i$$

Thus conditional joint probability distribution of $\vec{p}$ given $O$ can be written as:

$$P(\vec{p} \mid O) = \text{Dir}(v_1 + 1, v_2 + 1, \ldots, v_m + 1)$$
$$dp_m dp_{m-1} \ldots dp_2 \tag{8}$$

Now we can integrate the above equation over a subset of $(m-1)$-simplex, such that $p_1 > \max_{i=2}^m p_i$. This gives us the equation:

$$P(p_1 > \max_{i=2}^m p_i \mid O)$$
$$= \int_0^1 \int_{\mathcal{S}(p_1')} P(\vec{p} \mid O)$$
$$dp_2 \cdots dp_m dp_1',$$

where
$$\mathcal{S}(p_1') = \{(p_2, \ldots, p_m) \mid p_1' > \max_{i=2}^m p_i,$$
$$\sum_{i=2}^m p_i = 1 - p_1'\}. \tag{9}$$

We note that the integration has no closed-form solution, and we use numerical approximation to compute the above integral.

**Defining region of integration:** $\mathcal{S}(p_1')$  Next, for computation of Equation (9), we need to precisely calculate the limits of each integration such that they represent the region $\mathcal{S}(p_1')$. We do so by noting the following constraints on $p_i$: 1.) The $p_i = 0$ is valid $\forall 2 \leq i \leq m$, 2.) Given $\{p_m, p_{m-1} \ldots p_{i+1}$ are fixed and in region $\mathcal{S}(p_1')$, $p_i < \frac{1 - \sum_{j=i+1}^m p_j}{2}$ as else $p_i \geq p_1$ which is not allowed, 3.) Since $p_1 > \max_{j=i+1}^m p_j$, so $p_i < 1 - \sum_{j=i+1}^m p_j - \max_{j=i+1}^m p_j$, as else the $\vec{p}$, will lie outside the $(m-1)$-simplex, which is invalid. The first condition makes the lower limit for each integration 0, and the minimum of condition 2 and condition 3 gives the upper bound (limit) on each of the integrations.

**BETA stopping criteria** Due to $m - 1$ dimensions integrations involved, with $m$ often getting larger than 10, computing Equation (9) is not efficient. Instead, we observe that establishing the majority of $p_1$ over the next largest probability, $p_2$, is sufficient for our purpose. Then, pdf simplifies to BETA distribution with parameters $v_1 + 1, v_2 + 1$, and Equation (9) simplifies to:

$$\int_0^{0.5} p_2^{v2} \cdot (1 - p_2)^{v_1} dp_2 \qquad (10)$$

We use Scipy library in Python numerically compute the above equation.

| | | Self-Consistency | Adaptive-Consistency | | Δ | |
|---|---|---|---|---|---|---|
| | | Accuracy | Avg. Gen. | Accuracy | Gen. Reduc. | Acc. Diff. ↑ |
| **GSM-8K** | Vicuna-13B | $31.6_{(\pm 3.0)}$ | $26.8_{(\pm 2.2)}$ | $31.5_{(\pm 3.0)}$ | 1.4× | −0.1 |
| | Code-davinci-002 | $81.1_{(\pm 0.3)}$ | $13.8_{(\pm 0.0)}$ | $81.0_{(\pm 0.3)}$ | 2.9× | −0.1 |
| | GPT-3.5-turbo | **82.7** | **9.2** | 82.7 | 4.3× | 0.0 |
| **SVAMP** | Vicuna-13B | $63.0_{(\pm 0.3)}$ | $18.8_{(\pm 0.1)}$ | $62.8_{(\pm 0.4)}$ | 2.1× | −0.2 |
| | Code-davinci-002 | $85.1_{(\pm 0.3)}$ | $9.5_{(\pm 0.1)}$ | $85.0_{(\pm 0.3)}$ | 4.2× | −0.1 |
| | GPT-3.5-turbo | **85.1** | **9.5** | 85.0 | 4.2× | −0.1 |
| **ASDIV** | Vicuna-13B | $64.0_{(\pm 0.3)}$ | $16.5_{(\pm 0.2)}$ | $64.0_{(\pm 0.3)}$ | 2.4× | 0.0 |
| | Code-davinci-002 | $83.2_{(\pm 0.2)}$ | $10.0_{(\pm 0.0)}$ | $83.2_{(\pm 0.2)}$ | 4.0× | 0.0 |
| | GPT-3.5-turbo | **83.0** | **10.0** | 83.0 | 4.0× | 0.0 |
| **DATE UNDERSTANDING** | Vicuna-13B | $59.8_{(\pm 0.3)}$ | $17.3_{(\pm 0.3)}$ | $60.2_{(\pm 0.4)}$ | 2.3× | +0.4 |
| | Code-davinci-002 | $80.3_{(\pm 0.1)}$ | $10.7_{(\pm 0.3)}$ | $79.5_{(\pm 0.3)}$ | 3.7× | −0.8 |
| | GPT-3.5-turbo | **77.5** | **9.1** | 77.0 | 4.4× | −0.5 |
| **TRACKING SHUFFLED OBJECTS** | Vicuna-13B | $31.8_{(\pm 1.0)}$ | $20.3_{(\pm 0.0)}$ | $32.0_{(\pm 1.2)}$ | 2.0× | +0.2 |
| | Code-davinci-002 | $77.2_{(\pm 1.3)}$ | $9.7_{(\pm 0.1)}$ | $77.1_{(\pm 1.6)}$ | 4.1× | −0.1 |
| | GPT-3.5-turbo | 85.2 | **6.2** | 85.6 | 6.4× | +0.4 |
| **LOGICAL DEDUCTION** | Vicuna-13B | $51.2_{(\pm 0.8)}$ | $18.1_{(\pm 0.2)}$ | $51.4_{(\pm 0.6)}$ | 2.2× | +0.2 |
| | Code-davinci-002 | $89.4_{(\pm 0.2)}$ | $8.5_{(\pm 0.1)}$ | $89.4_{(\pm 0.2)}$ | 4.7× | 0.0 |
| | GPT-3.5-turbo | **86.8** | **7.5** | 86.8 | 5.3× | 0.0 |
| **STRATEGYQA** | Vicuna-13B | $65.8_{(\pm 0.5)}$ | $16.3_{(\pm 0.1)}$ | $65.8_{(\pm 0.4)}$ | 2.5× | 0.0 |
| | Code-davinci-002 | $79.0_{(\pm 0.2)}$ | $11.9_{(\pm 0.2)}$ | $78.8_{(\pm 0.1)}$ | 3.4× | −0.2 |
| | GPT-3.5-turbo | **68.1** | **11.8** | 67.9 | 3.4× | −0.2 |
| **BOOLEAN EXPRESSIONS** | Vicuna-13B | $79.2_{(\pm 0.6)}$ | $16.2_{(\pm 0.3)}$ | $78.4_{(\pm 0.3)}$ | 2.5× | −0.8 |
| | Code-davinci-002 | $94.5_{(\pm 0.4)}$ | $6.6_{(\pm 0.1)}$ | $94.5_{(\pm 0.4)}$ | 6.0× | 0.0 |
| | GPT-3.5-turbo | **93.2** | **5.0** | 92.8 | 7.9× | −0.4 |
| **SNARKS** | Vicuna-13B | $73.2_{(\pm 1.0)}$ | $23.2_{(\pm 0.7)}$ | $73.6_{(\pm 0.8)}$ | 1.7× | +0.4 |
| | Code-davinci-002 | $74.0_{(\pm 1.0)}$ | $12.7_{(\pm 0.4)}$ | $74.0_{(\pm 1.5)}$ | 3.1× | 0.0 |
| | GPT-3.5-turbo | **65.7** | **8.8** | 65.2 | 4.5× | −0.6 |
| **RUIN NAMES** | Vicuna-13B | $43.6_{(\pm 2.1)}$ | $33.8_{(\pm 0.6)}$ | $43.6_{(\pm 2.1)}$ | 1.2× | 0.0 |
| | Code-davinci-002 | $78.0_{(\pm 0.9)}$ | $17.2_{(\pm 0.1)}$ | $78.0_{(\pm 0.6)}$ | 2.3× | 0.0 |
| | GPT-3.5-turbo | **74.8** | **13.1** | 74.0 | 3.1× | −0.8 |
| **SALIENT TRANSLATION** | Vicuna-13B | $28.9_{(\pm 2.4)}$ | $28.7_{(\pm 2.5)}$ | $28.7_{(\pm 2.5)}$ | 1.2× | −0.3 |
| | Code-davinci-002 | $64.3_{(\pm 0.2)}$ | $11.8_{(\pm 0.5)}$ | $64.3_{(\pm 0.2)}$ | 3.4× | 0.0 |
| | GPT-3.5-turbo | **56.8** | **11.1** | 56.8 | 3.6× | 0.0 |
| **DISAMBIGUATION QA** | Vicuna-13B | $63.7_{(\pm 0.7)}$ | $22.8_{(\pm 1.0)}$ | $63.5_{(\pm 1.1)}$ | 1.8× | −0.3 |
| | Code-davinci-002 | $74.9_{(\pm 0.8)}$ | $13.5_{(\pm 0.6)}$ | $75.1_{(\pm 0.7)}$ | 3.0× | +0.1 |
| | GPT-3.5-turbo | **62.5** | **13.9** | 62.5 | 2.9× | 0.0 |
| **PENGUINS** | Vicuna-13B | $46.8_{(\pm 1.8)}$ | $22.9_{(\pm 0.7)}$ | $47.3_{(\pm 1.9)}$ | 1.7× | +0.5 |
| | Code-davinci-002 | $83.8_{(\pm 0.9)}$ | $11.0_{(\pm 0.4)}$ | $84.0_{(\pm 0.6)}$ | 3.6× | +0.2 |
| | GPT-3.5-turbo | **71.9** | **14.2** | 71.9 | 2.8× | 0.0 |
| **Average** | | $70.3_{(\pm 0.8)}$ | $14.1_{(\pm 0.5)}$ | $70.2_{(\pm 0.8)}$ | 3.2× | **-0.07** |

Table 5: Comparison of Adaptive-Consistency with Self-Consistency on 17 diverse coding & reasoning datasets. Self-Consistency always draws 40 samples. The table shows accuracy, average generations (Avg. Gen.). The Δ columns display reductions in generations (Gen. Reduc.) and accuracy (Acc. Diff.) between Self-Consistency and Adaptive-Consistency. Adaptive-Consistency achieves a 3.2× reduction in sample budget (*Gen. Reduc.*) with minimal average accuracy drop of 0.07% (*Acc. Diff.*).

| Dataset | Model | P-Value (Accuracy) | P-Value (Num Gens) |
|---|---|---|---|
| GSM-8K | VICUNA-13B | 0.5 | 0.0056 |
| GSM-8K | CODE-DAVINCI-002 | 0.42 | 2.09E-06 |
| SVAMP | VICUNA-13B | 0.07 | 3.47E-06 |
| SVAMP | CODE-DAVINCI-002 | 0.42 | 7.40E-06 |
| ASDIV | VICUNA-13B | 1 | 0.0005 |
| ASDIV | CODE-DAVINCI-002 | 1 | 0.0023 |
| DATE UNDERSTANDING | VICUNA-13B | 0.057 | 7.68E-05 |
| DATE UNDERSTANDING | CODE-DAVINCI-002 | 0.04 | 4.63E-05 |
| TRACKING SHUFFLED OBJECTS | VICUNA-13B | 0.5 | 0.00002 |
| TRACKING SHUFFLED OBJECTS | CODE-DAVINCI-002 | 0.67 | 9.88E-06 |
| LOGICAL DEDUCTION | VICUNA-13B | 0.5 | 0.0007 |
| LOGICAL DEDUCTION | CODE-DAVINCI-002 | - | 0.0016 |
| STRATEGYQA | VICUNA-13B | 0.90 | 1.16E-05 |
| STRATEGYQA | CODE-DAVINCI-002 | 0.24 | 0.0005 |
| BOOLEAN EXPRESSIONS | VICUNA-13B | 0.32 | 8.52E-05 |
| BOOLEAN EXPRESSIONS | CODE-DAVINCI-002 | - | 4.98E-06 |
| SNARKS | VICUNA-13B | 0.18 | 0.0007 |
| SNARKS | CODE-DAVINCI-002 | 1 | 0.0001 |
| RUIN NAMES | VICUNA-13B | - | 0.0049 |
| RUIN NAMES | CODE-DAVINCI-002 | 1 | 8.72E-06 |
| SALIENT TRANSLATION | VICUNA-13B | 0.18 | 0.0211 |
| SALIENT TRANSLATION | CODE-DAVINCI-002 | 1 | 0.0001 |
| DISAMBIGUATION QA | VICUNA-13B | 0.53 | 0.0015 |
| DISAMBIGUATION QA | CODE-DAVINCI-002 | 0.42 | 0.0002 |
| PENGUINS | VICUNA-13B | 0.18 | 0.0009 |
| PENGUINS | CODE-DAVINCI-002 | 0.42 | 7.79E-05 |
| **Average** | | 0.503 | 0.0002 |

Table 6: P-values using 2 sample t-test over 3 seeds on multiple datasets and models. The p-value for 'number of generations' is significantly less than 0.05 (average: 1.5e-3), confirming our method's efficiency, while the p-value for accuracy is much larger than 0.05 (average: 0.50), indicating that the slight accuracy difference is statistically insignificant.

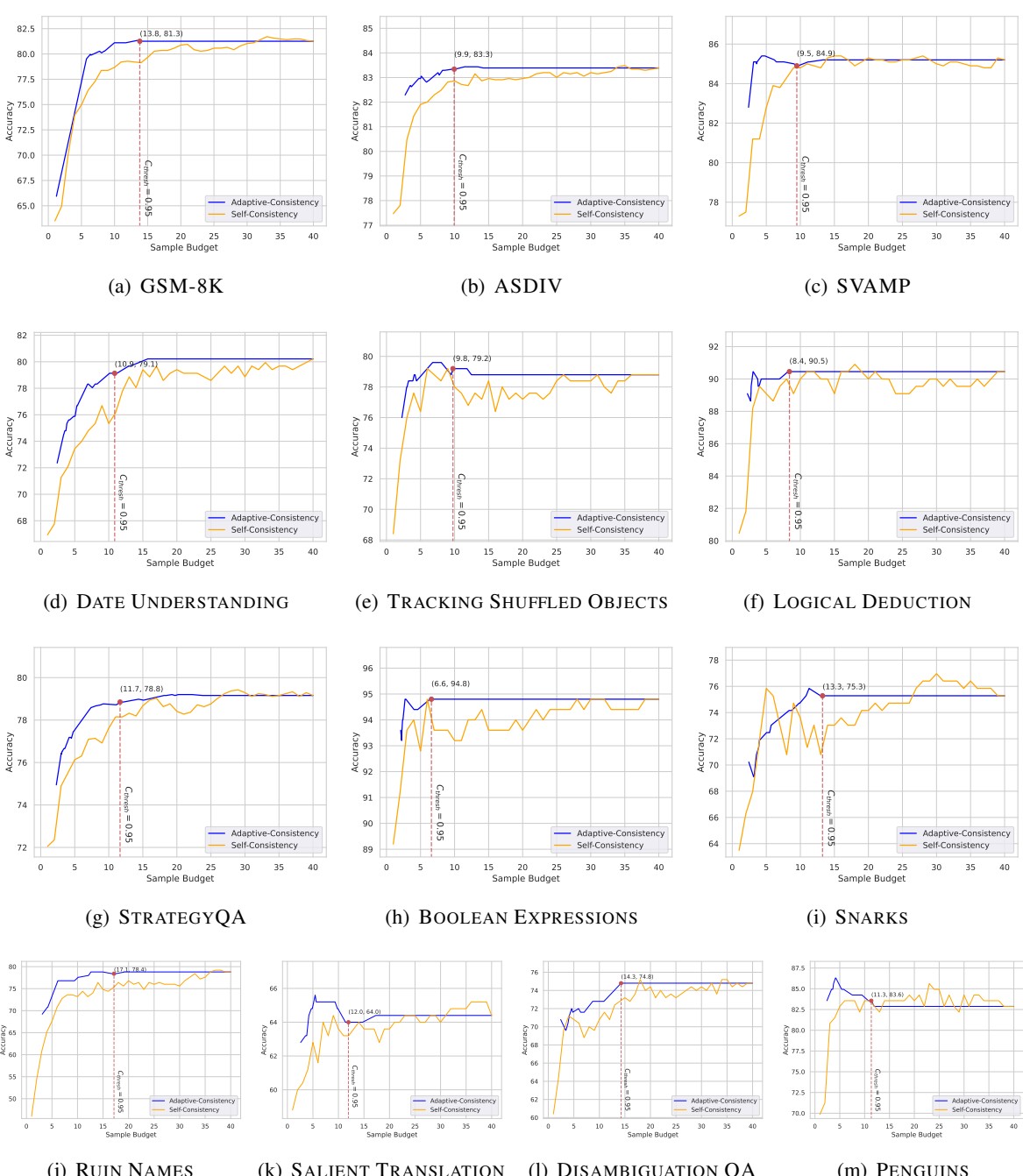

Figure 5: Comparison of Adaptive-Consistency with Self-Consistency on various average sampling costs. Adaptive-Consistency is able to consistently beat Self-Consistency, especially when the sampling cost is low. Moreover, $C_{thresh} = 0.95$ is a good indication of saturation in accuracy indicating the value works out-of-box for most configurations considered.

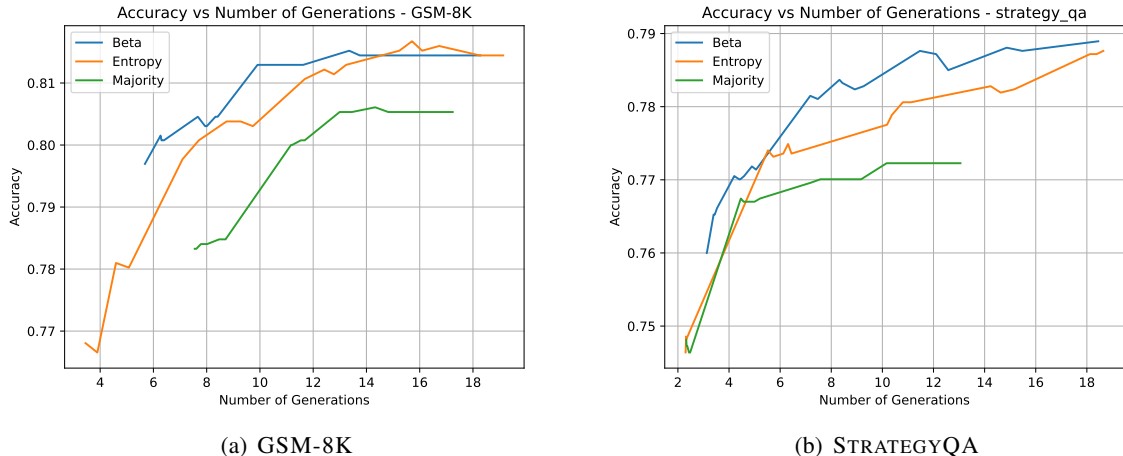

(a) GSM-8K  (b) STRATEGYQA

Figure 6: Comparison of BETA, MAJORITY and ENTROPY stopping criterias. In the two representative datasets, BETA consistently beats ENTROPY and MAJORITY for the same sampling cost. This shows in practice BETA performs better than both for the desirable range of accuracy and sampling cost.

| | | RANDOM | | MAJORITY | | ENTROPY | | BETA (Adaptive-Consistency) | | DIRICHLET | | CRP | |
|---|---|---|---|---|---|---|---|---|---|---|---|---|---|
| | | Average ↓ Generations | Accuracy ↑ | Average ↓ Generations | Accuracy ↑ | Average ↑ Generations | Accuracy ↑ | Average ↑ Generations | Accuracy ↑ | Average ↑ Generations | Accuracy ↑ | Average ↑ Generations | Accuracy ↑ |
| **GSM-8K** | VICUNA-13B | **26.0** | 30.1 | 28.7 | 31.5 | 26.3 | 31.5 | 26.8 | 31.5 | 28.2 | **31.7** | **25.6** | 31.5 |
| | CODE-DAVINCI-002 | 13.8 | 76.9 | 16.6 | 80.9 | 15.3 | 81.0 | 13.8 | 81.0 | 15.2 | **81.1** | **13.2** | 81.1 |
| **ASDIV** | VICUNA-13B | 28.0 | 63.2 | **14.8** | 63.7 | 15.8 | 63.9 | 16.5 | **64.0** | 17.7 | **64.0** | 16.9 | **64.0** |
| | CODE-DAVINCI-002 | 13.8 | 81.9 | **9.2** | 83.1 | 11.5 | **83.3** | 10.0 | 83.2 | 10.7 | 83.1 | 10.7 | 83.1 |
| **SVAMP** | VICUNA-13B | 28.0 | 61.3 | **17.1** | 62.5 | 17.7 | 62.6 | 18.8 | 62.8 | 19.7 | **62.9** | 18.2 | 62.8 |
| | CODE-DAVINCI-002 | 13.4 | 83.3 | **8.4** | 84.8 | 10.7 | 85.1 | 9.5 | 85.0 | 10.3 | **85.1** | 9.8 | 85.0 |
| **DATE UNDERSTANDING** | VICUNA-13B | 28.0 | 58.3 | **15.3** | 59.5 | 16.0 | 59.9 | 17.3 | **60.2** | 18.5 | 59.9 | 16.9 | 59.9 |
| | CODE-DAVINCI-002 | 13.2 | 76.4 | **9.7** | 78.7 | 11.6 | 79.9 | 10.7 | 79.5 | 11.9 | **80.5** | 10.7 | 79.8 |
| **TRACKING SHUFFLED OBJECTS** | VICUNA-13B | 27.9 | 31.8 | **15.0** | **33.0** | 18.4 | 32.0 | 20.3 | 32.0 | 23.3 | 32.0 | 19.6 | 31.8 |
| | CODE-DAVINCI-002 | 13.5 | 76.3 | **7.0** | 76.8 | 11.5 | 76.9 | 9.7 | 77.1 | 11.5 | **77.2** | 10.2 | 77.1 |
| **LOGICAL DEDUCTION** | VICUNA-13B | 27.9 | 50.5 | **12.9** | 51.2 | 15.8 | **51.4** | 18.1 | **51.4** | 20.9 | 51.2 | 18.3 | **51.4** |
| | CODE-DAVINCI-002 | 13.7 | 88.3 | **5.9** | **89.6** | 10.1 | **89.6** | 8.5 | 89.4 | 10.2 | 89.2 | 9.3 | 89.4 |
| **STRATEGYQA** | VICUNA-13B | 28.1 | 65.1 | **11.7** | 65.5 | 14.5 | 65.8 | 16.3 | 65.8 | 18.7 | **65.8** | 17.0 | 65.7 |
| | CODE-DAVINCI-002 | 13.4 | 76.6 | **7.2** | 77.8 | 14.9 | 78.5 | 11.9 | 78.8 | 14.5 | **78.9** | 11.4 | **78.9** |
| **BOOLEAN EXPRESSIONS** | VICUNA-13B | 27.6 | 78.0 | **10.4** | 76.8 | 14.8 | 78.3 | 16.2 | 78.4 | 19.1 | **78.8** | 17.0 | 78.5 |
| | CODE-DAVINCI-002 | 13.1 | 93.4 | **4.3** | 94.3 | 8.2 | **94.5** | 6.6 | **94.5** | 8.2 | **94.5** | 7.9 | 94.4 |
| **SNARKS** | VICUNA-13B | 28.4 | 70.3 | **18.1** | 72.1 | 20.3 | 73.0 | 23.2 | **73.6** | 25.8 | **73.6** | 22.9 | **73.6** |
| | CODE-DAVINCI-002 | 13.6 | 71.6 | **10.5** | **74.0** | 12.1 | **74.0** | 12.7 | **74.0** | 14.2 | 73.4 | 12.3 | 73.2 |
| **RUIN NAMES** | VICUNA-13B | **28.3** | 40.6 | 30.4 | 43.9 | 31.9 | 43.7 | 33.8 | 43.6 | 34.0 | 43.6 | 32.0 | **44.0** |
| | CODE-DAVINCI-002 | **13.8** | 71.7 | 17.5 | 77.7 | 18.6 | **78.1** | 17.2 | 78.0 | 17.6 | 76.8 | 16.4 | **78.1** |
| **SALIENT TRANSLATION** | VICUNA-13B | 24.9 | 27.7 | **24.6** | 28.5 | 26.3 | 28.0 | 28.7 | 28.7 | 29.4 | 28.8 | 26.9 | **28.9** |
| | CODE-DAVINCI-002 | 14.0 | 62.5 | **9.9** | **64.7** | 13.1 | 64.3 | 11.8 | 64.3 | 13.7 | 64.1 | 11.7 | 64.3 |
| **DISAMBIGUATION QA** | VICUNA-13B | 27.9 | 62.9 | **18.3** | 63.5 | 20.1 | 63.1 | 22.8 | 63.5 | 25.4 | **63.9** | 22.1 | 63.3 |
| | CODE-DAVINCI-002 | 13.7 | 72.1 | **10.4** | 73.9 | 15.9 | 74.9 | 13.5 | 75.1 | 16.3 | **75.2** | 13.2 | 75.2 |
| **PENGUINS** | VICUNA-13B | 27.9 | 45.6 | **19.7** | 46.3 | 20.7 | **47.3** | 22.9 | **47.3** | 25.1 | **47.3** | 22.1 | **47.3** |
| | CODE-DAVINCI-002 | 13.3 | 81.4 | **9.0** | 83.3 | 13.1 | 83.8 | 11.0 | 84.0 | 12.9 | 84.0 | 11.0 | **84.5** |

Table 7: Comparison of various Stopping Criterias in Adaptive-Consistency. In general, BETA outperforms RANDOM and MAJORITY by decent margins across all datasets. BETA has comparable performance to DIRICHLET, but the latter is much slower. ENTROPY performs similarly to BETA but lacks human-interpretable stopping rationale.

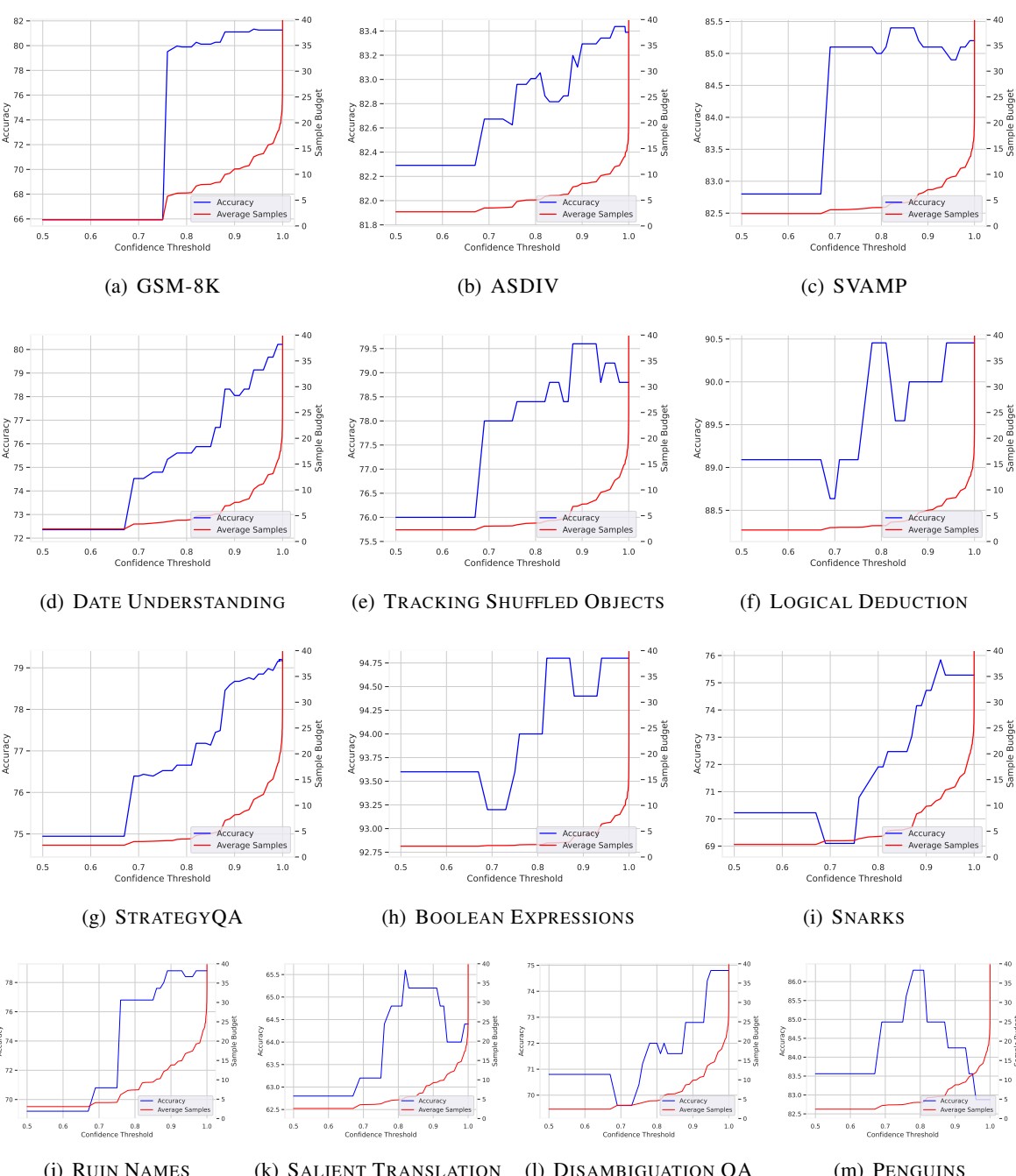

Figure 7: Impact of Confidence Threshold ($C_{thresh}$) on Adaptive-Consistency: As $C_{thresh}$ varies, the accuracy of Adaptive-Consistency increases gradually, eventually plateauing. Initially, the average number of generations also increases gradually but then sharply climbs, reflecting the accuracy-confidence trade-off. The trend is observed almost consistently across all datasets.