# OpenReview forum: "Let's Sample Step by Step: Adaptive-Consistency for Efficient Reasoning and Coding with LLMs"
_EMNLP/2023/Conference — EMNLP 2023 Main_

### Official Review · Reviewer_FcEp · 2023-07-30

**Soundness:** 4

**Excitement:**

4: Strong: This paper deepens the understanding of some phenomenon or lowers the barriers to an existing research direction.

**Paper Topic And Main Contributions:**

The paper proposes a technique called Adaptive-Consistency, which dynamically determines the number of samples
generated from LLM to draw the conclusion. The existing Self-consistency technique can improve the stability of the output
of LLMs by generating multiple output samples and selecting the most frequent one as the systems' output. While self-consistency
always needs a fixed number of samples. The proposed method can dynamically determine the number of samples and contribute to reducing the average number of samples. Experimental results show that the proposed method can significantly reduce the number of samples with
a negligible drop in accuracy with combinations of multiple datasets and LLMs.

**Reasons To Accept:**

1. The paper is very clearly written. I feel no difficulty in reading the paper.

2. The proposed method is simple but addresses an important problem. The proposed adaptive-consistency technique would be beneficial for a wide range of LLM applications.

3. Extensive evaluations clearly show the effectiveness of the proposed approach.

**Reasons To Reject:**

Although the extension for the code generation task and experimental evaluations are important and
specific to the LLM, the problem addressed in this paper is basically a statistical inference problem of
estimating parameters of the population distribution from samples, which is not specific to LLM, assuming that samples
are randomly generated. I suspect that basic statistical inference techniques based on confidence intervals can be
applied to this problem.

**Reproducibility:**

5: Could easily reproduce the results.

**Reviewer Confidence:**

3: Pretty sure, but there's a chance I missed something. Although I have a good feel for this area in general, I did not carefully check the paper's details, e.g., the math, experimental design, or novelty.

---

> ### Author Rebuttal · Authors · 2023-08-29
>
> Thank you for your thoughtful review of our paper! We are pleased to know that you found the paper clearly written and easy to comprehend. We appreciate your recognition of the simplicity yet effectiveness of the Adaptive-Consistency technique, and its potential benefits across various LLM applications.
>
> We think that all your questions are addressable within this discussion period. Please see our response below. We would love to address additional questions during the discussion period if anything is unclear.
>
> ---
>
> **The problem addressed in this paper is basically a statistical inference problem of estimating parameters of the population distribution from samples, which is not specific to LLM, assuming that samples are randomly generated**
>
> We agree with your observation that the proposed approach has general applicability and isn't confined to LLMs. As you rightly noted, its relevance extends to various online sampling scenarios, especially when sampling is resource-intensive and an early stopping mechanism is advantageous. The related work on crowdsourcing, as exemplified in Section 6, has explored nuanced solutions to the same challenge, and we draw inspiration from these techniques.
>
> ---
>
> **I suspect that basic statistical inference techniques based on confidence intervals can be applied to this problem.**
>
> Thank you for your suggestion regarding the applicability of confidence intervals to our problem. We aim to provide a probability that a particular element is the majority, rather than estimating a range for a parameter. Consequently, our methodology is tailored to offer a direct probability estimate (Equation 1 and Appendix D).
>
> To demonstrate the efficacy of Adaptive-Consistency, we tested various baselines, including majority voting, entropy, and CRP. As illustrated in Table 1 and Appendix Table 5, these methods either fall short of our approach in terms of performance or are notably resource-intensive.
>
> If the reviewer has specific suggestions on integrating confidence interval-based methods with our approach, we will be happy to explore them during the discussion period.

---

### Official Review · Reviewer_pwx4 · 2023-08-04

**Typos Grammar Style And Presentation Improvements:** L427
**Soundness:** 3

**Excitement:**

4: Strong: This paper deepens the understanding of some phenomenon or lowers the barriers to an existing research direction.

**Paper Topic And Main Contributions:**

This paper proposes Adaptive-Consistency as a more efficient alternative to Self-Consistency. The key idea is that, instead of making a fixed number of calls to an LLM and taking the majority as the answer (aka. Self-Consistency), the authors propose taking the posterior distribution under a Dirichlet-Multinomial model, and preemptively stop making calls to the LLM when one of the answers has over 95% probability of being the most likely answer (aka. their Dirichlet stopping criterion). They also explore variants of this approach which instead use a Chinese restaurant process (which models the fact that the next sample of the LLM may be completely different from the ones seen so far) and a simpler Beta-Bernoulli model with the two most frequent answers (which is more amenable to computation). They find that their approach reduces the number of calls made to an LLM by a factor of 2-4x, with essentially no impact in downstream performance.

**Questions For The Authors:**

Question A: Can you elaborate more on the setup for the dynamic budget experiments in section 5.2?

**Reasons To Accept:**

**This is a neat and intuitive idea, and the results are very encouraging**. This is an interesting application of more traditional probabilistic models to the field of LLMs, with results suggesting how a Self-Consistency—a common technique—can be substantially improved by preemptively stopping sampling when one can identify the most likely output with high probability.

**Practically useful.** These results indicate that there could be significant monetary cost (and maybe time) savings when repeatedly querying LLMs to get a consensus answer.

**Reasons To Reject:**

**Experiments with dynamic budget are not very clear**. It is not clear exactly what the setup in section 5.2 is. Do you fix a budget before hand (that is shared across all datapoints), and then run your approaches and the Self-Consistency baseline until the budget is depleted? If so, what would happen if the budget runs out early, e.g., before all datapoints are processed? I would expect to see more details about this on the paper, or in an appendix.

**It is not clear how expensive it is to perform the numerical integrations necessary for the Dirichlet/Beta/CRP models**. While these numbers are not critical from monetary cost perspective, they are important to understand whether this method also results in net wall time savings in getting an answer from an LLM.

**No explicit mention that this presupposes that one does not have access to LLM probabilities**. The way I see it, this approach is best suited for the setting where ones does not have access to the probabilities of outputs under the LLM (e.g., GPT-4). If one *did* have access to the LLM probabilities (e.g., Vicuna, one of the baselines), then this approach is less useful since one would never benefit from repeated observations of the same output. While this is not a limitation of the proposed method per se, I would expect at least some mention of this in the paper/limitations (maybe I missed it).

**Reproducibility:**

4: Could mostly reproduce the results, but there may be some variation because of sample variance or minor variations in their interpretation of the protocol or method.

**Reviewer Confidence:**

4: Quite sure. I tried to check the important points carefully. It's unlikely, though conceivable, that I missed something that should affect my ratings.

---

> ### Author Rebuttal · Authors · 2023-08-29
>
> Thank you for taking the time to review our paper! We were delighted to read that you found our Adaptive-Consistency approach both neat and intuitive, and appreciated our integration of traditional probabilistic models with LLMs. We're particularly encouraged that you recognized the practical implications, especially the potential for significant monetary and time savings when querying LLMs repeatedly.
>
> We think that all your questions are addressable within this discussion period. Please see our response below. We would love to address additional questions during the discussion period if anything is unclear.
>
> ---
>
> **Experiments with dynamic budget are not very clear.  / Question A: Can you elaborate more on the setup for the dynamic budget experiments in section 5.2?**
>
>
> Our primary objective is to compare the performance of Adaptive-Consistency (AC) and Self-Consistency (SC) across various sampling budgets. We first run AC with different thresholds (a lower threshold results in fewer generations, and vice-versa, as outlined in Section 5.1), producing a series of (#samples, performance) pairs for AC. Then, for each unique sample count from this set, we run SC to obtain a corresponding set of (# samples, performance) pairs for SC. Both sets of data points are presented in Figures 3 (detailed version in Figure 5.1), and show that AC consistently outperforms SC across all sampling costs. We will expand on this explanation and provide more examples in the next version to ensure clarity.
>
> ---
>
> **It is not clear how expensive it is to perform the numerical integrations necessary for the Dirichlet/Beta/CRP models**
>
> As shown in the table below, the Beta model (used in our experiments) takes less than 0.1 ms per sample, while the Dirichlet and CRP models average around 100ms. These times are *significantly less* than LLM inference, especially considering that our experiments were run on a single core machine.
>
> | Stopping Criteria | Time per sample (ms) |
> |-------------------|----------------------|
> | Beta              | 0.03                 |
> | Dirichlet        | 101.3                 |
> | CRP               | 94.6                  |
>
> We will include these details in the revised paper to clarify that the stopping criteria that are indeed much less expensive than LLM inferences.
>
>
> ---
>
> **No explicit mention that this presupposes that one does not have access to LLM probabilities**
>
> While our approach assumes no access to LLM probabilities, it's worth noting that Self-Consistency [1] has shown that even with access to these probabilities, multiple sampling outperforms a single pass. Further, several recent studies [2, 3] have demonstrated better performance using Self-Consistency despite having full access to model weights and token probabilities.
> We will include this discussion in the revised paper for clarity.
>
> ---
>
> **References**
> 1. Wang, X., Wei, J., Schuurmans, D., Le, Q., Chi, E.H., & Zhou, D. (2022). Self-Consistency Improves Chain of Thought Reasoning in Language Models. ArXiv, abs/2203.11171.
> 2. Li, Y., Choi, D.H., Chung, J., Kushman, N., Schrittwieser, J., Leblond, R., Tom, Eccles, Keeling, J., Gimeno, F., Lago, A.D., Hubert, T., Choy, P., de, C., d’Autume, M., Babuschkin, I., Chen, X., Huang, P., Welbl, J., Gowal, S., Alexey, Cherepanov, Molloy, J., Mankowitz, D.J., Robson, E.S., Kohli, P., de, N., Freitas, Kavukcuoglu, K., & Vinyals, O. (2022). Competition-level code generation with AlphaCode. Science, 378, 1092 - 1097.
> 3. Anil, Rohan et al. “PaLM 2 Technical Report.” ArXiv abs/2305.10403 (2023): n. pag.

---

### Official Review · Reviewer_tDcp · 2023-08-05

**Soundness:** 4

**Excitement:**

3: Ambivalent: It has merits (e.g., it reports state-of-the-art results, the idea is nice), but there are key weaknesses (e.g., it describes incremental work), and it can significantly benefit from another round of revision. However, I won't object to accepting it if my co-reviewers champion it.

**Paper Topic And Main Contributions:**

The authors present a straightforward and easy approach for improving decoding performance, essentially being self-consistency with an early exit based on a stopping criteria. They present results on a variety of tasks and current leading models.

**Questions For The Authors:**

In Section 5.2, how is the average number of samples enforced on Adaptive-Consistency? My (perhaps incorrect) understanding is that AC involves adding a stopping criteria, which does not provide direct control over the average.

**Reasons To Accept:**

The method is straightforward and well-justified, and the experiments are sufficiently comprehensive to demonstrate the effectiveness of this method (which mostly boils down to savings in number of samples over the "fixed-budget" self-consistency).

**Reasons To Reject:**

The exact analysis on performance vs budget is a little lighter than I would have liked (while acknowledging that there is a good amount of analysis already, and that these experiments are expensive to run given the large number of samples).
In particular, while Table 1 shows that Adaptive-Consistency achieves comparable performance with fewer samples, it is a little challenging to interpret the results without statistical tests or error bars.

**Reproducibility:**

4: Could mostly reproduce the results, but there may be some variation because of sample variance or minor variations in their interpretation of the protocol or method.

**Reviewer Confidence:**

4: Quite sure. I tried to check the important points carefully. It's unlikely, though conceivable, that I missed something that should affect my ratings.

---

> ### Author Rebuttal · Authors · 2023-08-29
>
> Thank you for taking the time to review our paper! We were happy to read that you found our method straightforward and well-justified, and that you appreciated our comprehensive experiments demonstrating the method's effectiveness, especially in terms of savings over the "fixed-budget" self-consistency.
>
> We think that all your questions are addressable within this discussion period. Please see our response below. We would love to address additional questions during the discussion period if anything is unclear.
>
> ---
>
> **In particular, while Table 1 shows that Adaptive-Consistency achieves comparable performance with fewer samples, it is a little challenging to interpret the results without statistical tests or error bars.**
>
>  Page 16-Table 4 shows the error bars calculated by evaluating Adaptive Consistency (our method) on 2 models with 3 different seeds. \
> Further, based on your suggestion, we have computed p-values for both accuracy and number of generations over 3 seeds using 2-sample t-test. The p-value for 'number of generations' is significantly less than 0.05 (average: 1.5e-3), confirming our method's efficiency, while the p-value for accuracy is much larger than 0.05 (average: 0.50), indicating that the slight accuracy difference is statistically insignificant.
> We will include these results with p-values and a detailed discussion in the revised paper.
> | Dataset             | Model   | P-Value (Accuracy) | P-Value (Num Gens) |
> |---------------------|---------|--------------------|--------------------|
> | gsm                 | Vicuna-13B  |                0.5 |     0.0056 |
> | gsm                 | Code-Davinci-002 (175B) |       0.42 |           2.09E-06 |
> | svamp               | Vicuna-13B  |      0.07 |           3.47E-06 |
> | svamp               | Code-Davinci-002 (175B) |       0.42 |           7.40E-06 |
> | asdiv               | Vicuna-13B  |                  1 |    0.0005  |
> | asdiv               | Code-Davinci-002 (175B) |                  1 |     0.0023 |
> | date                | Vicuna-13B  |      0.057 |           7.68E-05 |
> | date                | Code-Davinci-002 (175B) |      0.04   |           4.63E-05 |
> | tracking_three      | Vicuna-13B  |                0.5 |   0.00002  |
> | tracking_three      | Code-Davinci-002 (175B) |       0.67 |           9.88E-06 |
> | logical_three       | Vicuna-13B  |                0.5 |    0.0007  |
> | logical_three       | Code-Davinci-002 (175B) | -                  |     0.0016  |
> | strategy_qa         | Vicuna-13B  |       0.90   |           1.16E-05 |
> | strategy_qa         | Code-Davinci-002 (175B) |       0.24     |     0.0005   |
> | boolean_expressions | Vicuna-13B  |       0.32     |           8.52E-05 |
> | boolean_expressions | Code-Davinci-002 (175B) | -                  |           4.98E-06 |
> | snarks              | Vicuna-13B  |       0.18     |      0.0007  |
> | snarks              | Code-Davinci-002 (175B) |                  1 |    0.0001 |
> | ruin_names          | Vicuna-13B  | -                  |     0.0049   |
> | ruin_names          | Code-Davinci-002 (175B) |                  1 |           8.72E-06 |
> | salient_translation | Vicuna-13B  |       0.18    |      0.0211   |
> | salient_translation | Code-Davinci-002 (175B) |                  1 |    0.0001  |
> | disambiguation      | Vicuna-13B  |       0.53   |     0.0015  |
> | disambiguation      | Code-Davinci-002 (175B) |       0.42    |    0.0002  |
> | penguins            | Vicuna-13B  |       0.18     |    0.0009   |
> | penguins            | Code-Davinci-002 (175B) |       0.42    |           7.79E-05 |
> | average              |             |      0.503    |            0.0002    |
>
>
> ---
>
> **Q:  In Section 5.2, how is the average number of samples enforced on Adaptive-Consistency? My (perhaps incorrect) understanding is that AC involves adding a stopping criteria, which does not provide direct control over the average.**
>
> You are correct in noting that Adaptive-Consistency (AC) employs a threshold-based stopping criterion, and does not allow for directly controlling the number of generations. However, the confidence threshold in our stopping criterion influences the average number of samples: a lower threshold means AC stops earlier, possibly sacrificing accuracy, while a higher one uses more samples (Sec 5.1, L384). This threshold can be optimized using the training set, by adjusting it and observing the sample count.
>
> In Section 5.2, when we refer to the "average number of samples", we're presenting the empirical average observed across multiple instances. This observed average emerges as a result of AC's adaptive stopping — in some cases, it might stop early, while in others it might sample more, depending on the task. However, on average, AC consistently reduces the number of generations.

---

### Official Review · Reviewer_HiHf · 2023-08-11

**Soundness:** 4

**Excitement:**

4: Strong: This paper deepens the understanding of some phenomenon or lowers the barriers to an existing research direction.

**Paper Topic And Main Contributions:**

The paper introduces a new technique named Adaptive-Consistency, which dynamically adjusts the number of samples per question for improving the output of large language models (LLMs). Unlike traditional Self-Consistency methods that generate a constant number of samples, this approach non-uniformly distributes the sample budget according to the agreement level in the samples generated so far. The paper reports experimental results over 17 reasoning and code generation datasets and three LLMs, showcasing substantial improvements in sample budget efficiency with a minimal decrease in accuracy.

**Questions For The Authors:**

On what kind of tasks or datasets can adaptive consistency reduce more generations while maintaining comparable accuracy? Are there any commonalities in these datasets or tasks?

**Reasons To Accept:**

(1) Interesting Idea: The idea of dynamically adjusting the number of samples is innovative and addresses a significant limitation of existing Self-Consistency techniques. It has wide potential applications across different domains involving large language models.

(2) Efficiency: The claim that Adaptive-Consistency reduces the sample budget by up to 7.9 times is remarkable, and the minimal accuracy drop demonstrates the method's effectiveness and efficiency.

(3) Comprehensive Experiments: The paper's experimental design, covering 17 datasets and three different LLMs, provides broad insights into the method's performance across various scenarios.


**Reasons To Reject:**

No obvious reason to reject

**Reproducibility:**

4: Could mostly reproduce the results, but there may be some variation because of sample variance or minor variations in their interpretation of the protocol or method.

**Reviewer Confidence:**

4: Quite sure. I tried to check the important points carefully. It's unlikely, though conceivable, that I missed something that should affect my ratings.

---

> ### Author Rebuttal · Authors · 2023-08-29
>
> Thank you for your thoughtful review of our paper. We're pleased that you found value in our Adaptive-Consistency approach, appreciating its potential for dynamic sample adjustment in contrast to traditional Self-Consistency methods. We believe that all your inquiries can be addressed during this discussion period. Please see our response below. We would love to address additional questions during the discussion period if anything is unclear.
>
>
>
> ---
>
> **On what kind of tasks or datasets can adaptive consistency reduce more generations while maintaining comparable accuracy? Are there any commonalities in these datasets or tasks?**
>
> Adaptive-Consistency is particularly effective in two scenarios:
>
> 1. Tasks where a clear majority becomes apparent early during the sampling process. This allows Adaptive-Consistency to recognize dominant trends early on during the sampling process. For instance, as demonstrated in Figure 5c, on the 'Svamp' mathematical reasoning dataset, Adaptive-Consistency attains an accuracy similar to Self-Consistency, while utilizing an average of less than 5 samples per input.
>
> 2. Tasks that naturally produce a limited variety of unique answers. In such contexts, Adaptive-Consistency can efficiently deduce the majority due to the constrained answer space. For instance in 'boolean expressions' dataset, Adaptive Consistency reduces budget by 7.9 times, without hurting accuracy.

---

### Meta-Review · Area_Chair_eWEK · 2023-09-19

**Recommendation:** 5

**Metareview:**

The paper introduces the novel technique of Adaptive-Consistency for large language models (LLMs), addressing the limitation of traditional Self-Consistency by dynamically adjusting the number of samples. This adjustment is based on the agreement level in the generated samples, leading to more efficient and targeted sampling. Experimentally, across 17 reasoning and code generation datasets with three distinct LLMs, the technique exhibited an impressive reduction in sample budget (up to 7.9 times) with only a marginal compromise in accuracy. Reviewers noted simple but innovative approach, thorough experiments, and broad application potential.

Pros:

Innovative approach that dynamically adjusts sample numbers, addressing existing limitations in Self-Consistency.

Comprehensive experimental design and results across various datasets, showcasing efficiency and effectiveness.

Clear and well-justified presentation coupled with the potential for significant practical benefits in terms of cost and time.
Cons:

Some lack of clarity on experiments with a dynamic budget and potential complications if the budget is depleted prematurely.

Absence of details on the computational expense (in original draft) related to the numerical integrations for the Dirichlet/Beta/CRP models, raising questions about real-time efficiency.

---

### Decision · Program_Chairs · 2023-10-07

**Decision:**

Accept-Main

**Comment:**

The paper introduces the novel technique of Adaptive-Consistency for large language models (LLMs), addressing the limitation of traditional Self-Consistency by dynamically adjusting the number of samples. This adjustment is based on the agreement level in the generated samples, leading to more efficient and targeted sampling. Experimentally, across 17 reasoning and code generation datasets with three distinct LLMs, the technique exhibited an impressive reduction in sample budget (up to 7.9 times) with only a marginal compromise in accuracy. Reviewers noted simple but innovative approach, thorough experiments, and broad application potential.

Pros:

Innovative approach that dynamically adjusts sample numbers, addressing existing limitations in Self-Consistency.

Comprehensive experimental design and results across various datasets, showcasing efficiency and effectiveness.

Clear and well-justified presentation coupled with the potential for significant practical benefits in terms of cost and time.
Cons:

Some lack of clarity on experiments with a dynamic budget and potential complications if the budget is depleted prematurely.

Absence of details on the computational expense (in original draft) related to the numerical integrations for the Dirichlet/Beta/CRP models, raising questions about real-time efficiency.